# Intestinal Transcriptomic and Histologic Profiling Reveals Tissue Repair Mechanisms Underlying Resistance to the Parasite *Ceratonova shasta*

**DOI:** 10.3390/pathogens10091179

**Published:** 2021-09-13

**Authors:** Damien E. Barrett, Itziar Estensoro, Ariadna Sitjà-Bobadilla, Jerri L. Bartholomew

**Affiliations:** 1Department of Microbiology, Oregon State University, Corvallis, OR 97331-3804, USA; debarret@olemiss.edu; 2Fish Pathology Group, Instituto de Acuicultura Torre de la Sal, Consejo Superior de Investigaciones Científicas, 12595 Castellón, Spain; itziar.estensoro@csic.es (I.E.); ariadna.sitja@csic.es (A.S.-B.)

**Keywords:** *Oncorhynchus mykiss*, steelhead, *Ceratonova shasta*, myxozoan, parasites, RNA-seq, transcriptomics, immune response, host–parasite interaction

## Abstract

Background: Myxozoan parasites infect fish worldwide causing significant disease or death in many economically important fish species, including rainbow trout and steelhead trout (*Oncorhynchus mykiss*). The myxozoan *Ceratonova shasta* is a parasite of salmon and trout that causes ceratomyxosis, a disease characterized by severe inflammation in the intestine resulting in hemorrhaging and necrosis. Populations of *O. mykiss* that are genetically fixed for resistance or susceptibility to ceratomyxosis exist naturally, offering a tractable system for studying the immune response to myxozoans. The aim of this study was to understand how steelhead trout that are resistant to the disease respond to *C. shasta* once it has become established in the intestine and identify potential mechanisms of resistance. Results: Sequencing of intestinal mRNA from resistant steelhead trout with severe *C. shasta* infections identified 417 genes differentially expressed during the initial stage of the infection compared to uninfected control fish. A strong induction of interferon-gamma and interferon-stimulated genes was evident, along with genes involved in cell adhesion and migration. A total of 11,984 genes were differentially expressed during the late stage of the infection, most notably interferon-gamma, interleukin-6, and immunoglobulin transcripts. A distinct hardening of the intestinal tissue and a strong inflammatory reaction in the intestinal submucosa including severe hyperplasia and inflammatory cell infiltrates were observed in response to the infection. The massive upregulation of caspase-14 early in the infection, a protein involved in keratinocyte differentiation might reflect the rapid onset of epithelial repair mechanisms, and the collagenous stratum compactum seemed to limit the spread of *C. shasta* within the intestinal layers. These observations could explain the ability of resistant fish to eventually recover from the infection. Conclusions: Our results suggest that resistance to ceratomyxosis involves both a rapid induction of key immune factors and a tissue response that limits the spread of the parasite and the subsequent tissue damage. These results improve our understanding of the myxozoan–host dialogue and provide a framework for future studies investigating the infection dynamics of *C. shasta* and other myxozoans.

## 1. Introduction

Myxozoans are parasitic cnidarians characterized by a complex two-host, two-spore lifecycle that alternates between a myxospore stage that infects the invertebrate host (annelids or bryozoans) and an actinospore stage that infects the vertebrate host (generally fish) [1]. Myxozoans are widely distributed with over 2400 known species and have adapted to both freshwater and marine hosts [2]. Infections in the fish host are often asymptomatic, however, certain myxozoan species are known to cause severe pathology or death of the fish host. Mortality and morbidity as a result of myxozoan infections is becoming more common as aquaculture continues to expand worldwide, and more fish species become intensively managed by humans [3]. For example, throughout Asia and the Mediterranean, cultivation of turbot (*Scophthalmus maximus*) and several perciform fish is severely limited by enteromyxosis, caused by the myxozoan parasites *Enteromyxum scophthalmi* and *E. leei*, respectively [4,5]. Similarly, pharyngeal myxobolosis caused by *Myxobolus hunghuensis* is a limiting factor in the development of gibel carp (*Carassius auratus gibelio)* aquaculture in China [6]. Globally, over 35 species of marine fish are affected by *Kudoa thyrsites*, which encysts in muscle and causes postmortem myoliquefaction of the surrounding tissue, resulting in large economic losses due to poor fillet quality [7].

Rainbow trout and its anadromous form, steelhead trout (*Oncorhynchus mykiss*), are among the most widely cultivated species of fish and are negatively affected by several myxozoan pathogens, including *K. thyrsites*, *Myxobolus cerebralis* (the causative agent of whirling disease) [8], and *Tetracapsuloides bryosalmonae,* which is a related malacosporean and causes proliferative kidney disease [9]. Within the Pacific Northwest of the United States and Canada, *O. mykiss* and related salmonids are severely impacted by the intestinal myxozoan *Ceratonova shasta* (syn. *Ceratomyxa shasta*) which causes ceratomyxosis, a disease characterized by hemorrhaging and necrosis of the intestine and potentially death of the fish host. *C. shasta* has been linked to population level declines of Pacific Northwest wild fish stocks [10,11], and exerts such a selective pressure on the fish host that endemic populations became genetically fixed for resistance to the disease [12,13,14,15,16,17]. However, the parasite is not established in all river systems where salmonids are native and allopatric salmonids are highly susceptible to the parasite, and exposure to a single spore is capable of causing ceratomyxosis and mortality [18,19]. This leads to an almost binary resistance phenotype, where mortality either occurs after exposure to one spore (susceptible hosts), or thousands (resistant hosts) [11,20].

The combination of a highly virulent myxozoan, along with naturally occurring resistant and susceptible strains of the fish host, offers an attractive system for studying the immune response to myxozoan parasites. Our research group previously conducted a comparative transcriptomic analysis of resistant and susceptible phenotypes of steelhead trout exposed to a low dose of *C. shasta*, sufficient to cause mortality in the susceptible phenotype [21]. RNA-seq analysis revealed a downregulation of genes involved in the interferon-gamma (IFN-gamma) signaling pathway in the gills, the site of initial parasite invasion, of both phenotypes at 1 day post exposure (dpe) to *C. shasta*. By 7 dpe, resistant fish had effectively contained the infection, having either low or undetectable levels of *C. shasta* in their intestine, with several immune genes upregulated. In contrast to the resistant fish, susceptible fish had a significant parasite burden in their intestine, but no genes with known immune functions were upregulated, highlighting a lack of early recognition in these fish. The parasite continued to replicate exponentially in susceptible fish, and sequencing of intestinal mRNA from these fish at 14 and 21 dpe revealed an interferon-gamma driven T_H_1 response and a suppression of the T_H_17 response. This intense T_H_1 response failed to ameliorate the progress of the disease and the intestines of these fish progressively deteriorated. A comparable dataset from resistant fish with a similar parasite burden in their intestine could help us discern why the susceptible fishes’ immune response failed to offer any protection.

To understand how fish with a resistant phenotype respond to the parasite once it becomes established in the intestine, we challenged resistant steelhead trout with a dose of *C. shasta* sufficient to cause pathology. Accomplishing this task in the laboratory presented a significant hurdle given the difficulty of generating large quantities of actinospores (the infective parasite stage for the fish) and the extremely high resistance threshold of these fish. For this reason, we chose to focus on two key timepoints in the infection, representing the early and late immune response. We anticipated that these fish would respond with a stronger transcriptomic signal due to the higher parasite burden and that the early timepoint would confirm the results of our previous study: that resistant fish mount an earlier immune response to the parasite. Additionally, we believed it would provide a clear view of the early innate immune response, undistorted by the changes in gene expression that arises later in the infection due to the breakdown of the intestinal structure. The late timepoint would help characterize the adaptive immune response, as well as provide a dataset for comparison with susceptible fish from our previous study. We established severe infections by continuously exposing resistant steelhead trout to a high concentration of *C. shasta* for 5 days to achieve a high parasite burden in the intestine. Intestines were collected at 7 and 21 dpe and changes in gene expression were analyzed by RNA-seq, and the histopathological response was also investigated at both timepoints.

## 2. Results 

### 2.1. Infection of Resistant Steelhead Trout

The first clinical sign of *C. shasta* infection occurred at 19 dpe, when all the exposed fish stopped feeding, and thus anorexia became evident. At the 21 dpe sampling timepoint, the intestines of all the exposed fish were grossly swollen, with an apparent increase in capillarization and a hemorrhagic appearance. The tissue was rigid and difficult to cut and later homogenize in liquid nitrogen. Of the three exposed fish that remained after the 21 dpe sampling timepoint, one succumbed to the infection at 32 dpe. Its intestine appeared bloody and swollen and a vent swab revealed abundant mature myxospores. The remaining two fish resumed feeding again at 37 dpe and continued to do so until they were euthanized at 60 dpe. The parasite burden of the exposed fish was quantified by qPCR. The prevalence of infection was 100% among the exposed fish and at 7 dpe they had an average Cq of 27.0 ± 2.0, which increased to 15.2 ± 1.6 at 21 dpe (Figure 1). For reference, the 1 and 1000 actinospore standards have Cq values of 34 and 24, respectively. All of the control fish were negative by qPCR.

### 2.2. Histopathology

Histological observation evidenced a substantial hyperplasia in the *lamina propria*-submucosa of the infected fish at 21 dpe, which was not observed in control fish (Figure 2). At 7 dpe, intestines of infected fish maintained their basic tissue structure, though cell proliferation and infiltration in the submucosa were patent and localized detachment of the epithelium from the *lamina propria* occurred. These signs were significantly magnified in the intestines of infected fish at 21 dpe, which presented large areas of detached epithelia and a severe submucosal hyperplasia. At this timepoint, trophozoites as well as disporoblasts with mature spores were observed invading the submucosa, but they did not reach beyond the *stratum compactum* (Figure 3). This collagen sheath, which lies beneath the *lamina propria*, seemed to act as a barrier, preventing parasite from spreading into deeper layers. Inflammatory cell foci in infected fish mainly consisted of lymphocyte-like cells in the submucosa and infiltrating the epithelium, and granulocytes. The latter are expected to be eosinophils, though eosinophilia was very weakly distinguished in tissue sections due to fixation with Dietrich’s solution.

Interestingly, the fish that survived until 60 dpe were able to largely regenerate their intestinal structure. There was evidence of some localized epithelial detachment (Figure 2d) and large clusters of granulocytes were present along the submucosa (Figure 3d). However, the parasite was not completely cleared by 60 dpe, and *C. shasta* stages were still found, mainly in the intestinal lumen (Figure 2e).

A pattern of increasing goblet cell abundance with increasing time post-exposure was observed in both experimental groups. These goblet cells contained neutral mucins (PAS+) (Figure 3).

### 2.3. Sequencing of Intestinal mRNA

The sequencing of intestinal mRNA from resistant fish at 7 and 21 dpe generated 519 million single-end reads. The percentage of reads that could be uniquely mapped to a single locus in the rainbow trout genome varied between groups, ranging from 40.4% to 75.4% (Table 1). Principal component analysis showed a high degree of uniformity between samples within a group and clearly separated infected fish from uninfected controls with the first principal component explaining 63% of the total variation (Figure 4). Hierarchical clustering also clearly separated the infected samples from 21 dpe (Figure 5) and all the samples showed a high degree of correlation (r > 0.96), suggesting no outliers were present.

### 2.4. Differential Gene Expression in the Intestine at 7 dpe and Analysis of Key Gene Expression

The sequencing reads from 7 dpe were mapped to 32,959 genes in the rainbow trout genome, and 417 differentially expressed genes (DEGs) were identified in infected fish relative to their time-matched controls, including 338 (81%) upregulated genes and 78 (19%) downregulated genes (Figure 6 and Appendix A). Among the DEGs, 30 genes were strongly upregulated (Log2-FC > 4), including C-C motif chemokine 4-like (285.6 fold), two paralogs of C-C motif chemokine 13-like (46.7, 79.9 fold), three copies of C-C motif chemokine 19-like (27.8, 42.7, 51.4 fold), two paralogs of caspase-14-like (34.7, 139.2 fold), five paralogs of interferon-induced protein 44-like (20.6, 42.8, 68.7, 82.8, 86.4, 161.2 fold), interferon-gamma 2 precursor (32.4 fold), and cathelicidin 1 precursor (38.1 fold). Five genes were found to be strongly downregulated: protein CREG1-like (−969.7 fold), troponin T-fast skeletal muscle isoforms (−78.5 fold), cytochrome P450 2K4 (−38.4 fold), C-X-C motif chemokine 13-like (−29.2 fold), CUGBP Elav-like family member 3-B (−23.8 fold), and guanylyl cyclase-activating protein 2-like (−19.6 fold).

Based on the DEGs functional description and their associated GO terms, 114 DEGs known to have functions involved in immune responses were grouped into ten categories and presented in Table 2.

### 2.5. GO Enrichment 7 dpe

GO enrichment analysis was conducted on the 338 upregulated genes to gain insight into the biological functions of these DEGs, as well as the involved molecular processes. This identified 115 significantly enriched GO terms among the upregulated genes. ClueGo analysis clustered these terms into networks revolving around GO terms for “defense response to other organism”, “defense response”, “innate immune response”, and “interferon gamma-mediated signaling pathway” (Figure 7). No specific GO enrichment was found among the 78 downregulated genes.

### 2.6. Differential Gene Expression in the Intestine at 21 dpe and Analysis of Key Genes

The sequencing reads from 21 dpe mapped to 34,286 genes and 11,984 DEGs were identified in infected fish relative to their time-matched controls. Of these, 5801 (48%) were upregulated and 6183 (52%) were downregulated. In addition to the increased number of DEGs at 21 dpe, the magnitude of the differential expression also greatly increased, with 1,149 DEGs being strongly upregulated (Log2-FC > 4) (Figure 8 and Appendix A).

The most highly upregulated genes were those related to the immune system, cell communication, and tissue remodeling: interleukin-6-like (7118.6 fold), two paralogs of gap junction C × 32.2 protein-like (3516.0 and 2584.8 fold), chemokine CXCL13 precursor (3258.9 fold), interleukin-6 precursor (1851.8 fold), interleukin-10 precursor (1393.7 fold), fibroblast growth factor 21 (967.4 fold) and interleukin-1 beta-like (965.1 fold). The most significantly downregulated genes were primarily involved in metabolic pathways and transport of ions and solutes, and a few immune genes: solute carrier family 13 member 3-like (−430.3 fold), beta,beta-carotene 9′,10′-oxygenase-like (−323.4 fold), complement factor B (-276.2 fold), aquaporin-8-like (−176.4 fold), interferon-induced GTP-binding protein Mx2-like (−171.6 fold), and gastrotropin-like (−163.7 fold).

### 2.7. GO Enrichment 21 dpe

GO analysis of the 5801 upregulated genes revealed that they were enriched for 1135 GO terms, predominantly immune-related. ClueGO analysis clustered the biological process GO terms into five networks revolving around the terms “transmembrane receptor protein tyrosine kinase signaling pathway”, “immune response-activating signal transduction”, “cytokine secretion”, “immune response-activating cell surface receptor signaling pathway”, “T cell differentiation”, regulation of leukocyte migration”, and “positive regulation of actin filament polymerization” (Figure 9).

The same analysis of the 6189 downregulated genes found 491 enriched GO terms which ClueGO analysis clustered into networks based on metabolic and energy producing GO terms, including “generation of precursor metabolites and energy”, and “carboxylic acid catabolic process” (Figure 10).

### 2.8. Analysis of Key Immune Genes Differentially Expressed at 21 dpe

Based on the GO enrichment profiles and the most highly upregulated genes, a set of 89 key genes related to the innate and adaptive immune response that were differentially expressed in infected steelhead trout at 21 dpe were identified and are presented in Table 3.

### 2.9. Comparison to the Transcriptome of Susceptible Fish at 21 dpe

To understand the difference between the response of resistant fish and their susceptible counterparts, we compared the differential gene expression results from 21 dpe in this study to those previously generated by our research group for susceptible fish with a similar parasite burden at 21 dpe [21]. The experimental conditions of this study were designed to align with those of our previous one, and while other factors that influence gene expression cannot be ruled out, we believe this comparison will provide insight into the broader details of how resistant fish respond compared to their susceptible counterparts. Comparison of the 11,984 DEGs identified in resistant fish at 21 dpe with the DEGs found in susceptible fish at 21 dpe identified 7820 genes differentially expressed in both phenotypes (Figure 11). 28 of the 7820 were expressed in opposite directions in resistant fish compared to susceptible fish (supplemental). Among the 28 genes, 2 paralogs of “60 kDa heat shock protein, mitochondrial” were the only genes with known immune functions and were downregulated in resistant fish (−2.3 and −2.5-fold). The overall expression of the 7820 shared DEGs was highly positively correlated (r = 0.94, *p* < 2.2 × 10^−16^), and the average difference in fold change was 1.76 ± 1.73. GO enrichment analysis of the shared DEGs revealed 453 enriched GO terms, primarily related to metabolic processes, innate immune response, and cell adhesion. Analysis of the genes that were differentially expressed only in resistant fish revealed a similar profile with 302 enriched GO terms. However, adaptive immune system processes were much more prominent, including many terms that were not present among the shared DEGs or those unique to susceptible fish. These included “adaptive immune response”, “adaptive immune response based on somatic recombination of immune receptors built from immunoglobulin superfamily domains”, “lymphocyte mediated immunity”, and “T cell mediated immunity.”

The number of secreted IgT and IgM transcripts was quantified for both resistant and susceptible fish at 21 dpe to compare the antibody response between the two phenotypes. This revealed a much stronger induction of both immunoglobulins in resistant fish, particularly IgT, which was upregulated over 300-fold (Table 4).

## 3. Discussion 

The dialogue between host and parasite is complex, involving not only the immune system, but also changes in the host’s metabolism and the cellular structure of the infected tissues. RNA-seq offers a broad, non-targeted approach to understand this interaction, and can identify putative resistance genes for future functional studies. *Ceratonova shasta* offers an attractive biological model for studying the host-parasite dynamic, given that it infects well-studied and economically important salmonids that naturally occur on the phenotypic extremes of resistance. In this study, we sequenced intestinal RNA from resistant steelhead trout that were exposed to a high dose of *C. shasta* to elicit a vigorous immune response. Intestinal samples from 7 and 21 dpe were sequenced to further our understanding of the innate and adaptive immune response to this parasite. We found that by 7 dpe resistant fish were already mounting a vigorous IFN-gamma driven T_H_1 response that was accompanied by remodeling of the intestinal tissues. This continued at 21 dpe, where signs of a strong antibody response were evident along with a possible transition to a T_H_2 response. Despite the overwhelming parasite dose (>30,000 spores) these fish received and the damage to the intestinal structure that resulted, some fish were able to regenerate their intestinal tissues and resume feeding as normal. 

In our previous study comparing the transcriptomic response of resistant and susceptible steelhead trout exposed to a moderate dose of *C. shasta*, we observed that despite having a significant parasite burden in their intestine by 7 dpe, susceptible fish did not respond to the infection [21]. Although the exposure was not sufficient to establish an active infection in the intestines of the resistant fish in that study, we observed an upregulation of several immune genes in the intestine at 7 dpe, suggesting that a more rapid immune response was occurring in resistant fish. The results presented here confirm this, with resistant fish mounting a vigorous immune response at 7 dpe, with 338 genes upregulated. This response was predominantly an IFN-gamma driven T_H_1 response, with upregulation of numerous interferon-stimulated genes, as well as MHC class I genes and those involved in antigen processing and presentation. This would suggest that an adaptive immune response is already being mounted against the parasite. We also observed upregulation of NLRC5 and GTPase IMAP family member 4 (GIMAP 4). NLRC5 is a cytosolic pathogen recognition receptor involved in MHC class I-dependent immune responses [22] and was found to be upregulated in the gills and intestine of resistant fish in our previous study. In that same study, two paralogs of GIMAP 4, a protein involved in T-lymphocyte development [23], were the most highly upregulated genes in the gills of resistant fish at 1 dpe. Intriguingly, the exact same paralogs were the most highly upregulated immune genes in the intestine of susceptible fish at 14 dpe. The early induction of these genes in resistant fish compared to susceptible fish strongly suggests a role for these genes in *C. shasta* resistance. Although different paralogs of NLRC5 and GIMAP 4 were identified in this study, our findings further support their involvement in *C. shasta* resistance.

It is also important to highlight that despite being exposed to much higher concentration of *C. shasta* actinospores, and over a longer period of time, 2-fold less parasite DNA could be detected in the intestine of resistant fish at 7 dpe (Cq of 27.0 ± 2.0 for resistant fish vs. a Cq of 24.8 ± 0.8 for susceptible fish). This further supports the hypothesis that resistant fish can recognize the parasite and elicit an immune response early in the infection that is capable of either directly killing the parasite or inhibiting its replication. It remains to be determined precisely when and where this occurs, but it may be occurring in the blood vessels during *C. shasta* migration from the gills to the intestine. When resistant and susceptible fish have been exposed in parallel to *C. shasta*, no difference in parasite burden has been detected in the gills, and both phenotypes had a suppressed IFN-gamma response in this tissue [21,24]. Elimination of the parasite in the intestine does not seem to occur until much later in the infection (25+ dpe), and between 7 and 21 dpe the amount of parasite DNA in the intestine seems to follow an exponential growth curve in both resistant and susceptible fish. Our observations suggest that initial resistance either occurs in the blood or when the parasite first reaches the intestine.

While it was not surprising to find a strong induction of IFN-gamma, something that is commonly seen in *C. shasta* infected salmonids and other myxozoan infected fish [24,25,26,27], it was perhaps surprising that the early immune response was so strongly focused on this one pathway. IFN-gamma is the signature T_H_1 cytokine and mediates the response to viruses and other intracellular pathogens. *C. shasta* is considered an extracellular pathogen and certainly occupies that role in the intestine. This raises the question as to how or why a pathway geared towards intracellular pathogens is invoked and why a cytosolic pathogen recognition receptor (NLRC5) would be upregulated. This may be explained by *C. shasta* having an intracellular phase early in the infection. The only evidence for this comes from a study by Bjork & Bartholomew 2010, which reported the invasion and migration of fluorescently stained *C. shasta* [28]. The authors noted a potential intracellular phase in the endothelium of blood vessels starting at 3 dpe and suggested the use of electron microscopy to resolve the question of an intracellular developmental stage of the parasite. This would not be without precedent, as other myxozoans are known to have an early intracellular developmental stage or even having an intracellular final location. *M. cerebralis* initially replicates intracellularly in the epithelia of the epidermis of trout [29], and *Sphaerospora molnari* is intracellular in the gills of carp, prior to extracellular proliferation in the blood [30], and *Kudoa* species develop within muscle fibers. In the latter case, it has been suggested that the parasite remains undetected by the host immune system due to their intracellular location, firmly enveloped by the remnants of the muscle fiber and the surrounding connective network of the endomysium [31].

Whether an early intracellular phase is required for parasite development or represents a form of immune evasion remains to be determined. An intriguing possibility, at least for *C. shasta*, is that going intracellular causes the host to initiate a cytotoxic T_H_1 response that is ineffective against the extracellular stages in the intestine. This is evidenced by the upregulation of antigen presentation machinery and MHC class I molecules, which are responsible for the presentation of cytosol-derived peptides to CD8^+^ cytotoxic T lymphocytes (CTLs). Activated CTLs recognize and directly kill infected host cells, which is vital for controlling infections caused by intracellular pathogens [32]. A mistargeted immune response would explain why the rapid induction of IFN-gamma in the intestine at 7 dpe failed to slow parasite proliferation, with significantly more parasite DNA being detected at 21 dpe. 

If IFN-gamma is playing a role in resistance to *C. shasta* in the intestine, the actual mechanism remains unclear. IFN-gamma mediates parasite clearance by inducing the production of nitric oxide (NO) in classically activated macrophages [33,34]. This antiparasitic effect has been demonstrated for numerous parasites, including the well-studied intestinal parasites *Entamoeba histolytica*, *Cryptosporidium parvum*, and *Giardia spp.* [34,35,36]. In the present study, we did not observe upregulation of inducible nitric oxide synthase, which is responsible for NO production in macrophages, and instead observed upregulation of arginase at both 7 and 21 dpe. Arginase, a marker for alternatively activated macrophages, utilizes the same substrate as nitric oxide synthase (L-arginine) and redirects it to the formation of polyamines and proline, which are necessary for collagen synthesis and wound healing [37]. In addition to depleting the necessary substrate for NO production, upregulation of arginase also inhibits the expression of inducible nitric oxide synthase. It is also curious that we observed upregulation of arginase at 7 dpe in the absence of any T_H_2 cytokines, which typically drive its expression. The protozoan parasites *Trypanosoma brucei* and *T. cruzi* have both been shown to induce host expression of arginase in a T_H_2-cytokine independent manner as a means of suppressing NO production [38,39]. Our observations may indicate either a protective host response geared towards controlling inflammation and maintaining tissue integrity, or a pathogenic strategy to escape IFN-γ driven NO production. Given how important the interplay between classically activated NO producing macrophages and alternatively activated arginase expressing macrophages are to the resolution of parasitic infections, an in-depth analysis of the macrophage populations during *C. shasta* infection would greatly benefit our understanding of the host immune response.

In addition to developing an earlier immune response, resistant fish have a much different reaction to the parasite at the tissue level. At 7 dpe, evidence of tissue remodeling is present with the upregulation of junctional proteins, arginase, and the massive upregulation of caspase-14, which plays a terminal role in keratinocyte differentiation [40]. Keratins are cytoskeletal proteins classically used as epithelial cell markers during injury and disease in vertebrates [41,42] and which have also been described in non-cornified mucosal epithelia of teleosts [43]. In mice, keratinocyte growth factor has been shown to ameliorate drug-induced intestinal mucosal disruption through induction of epithelial repair and tight junction protein expression, together with the inhibition of increased epithelial permeability [44]. The early changes in gene expression observed in this study, including the upregulation of caspase-14, point towards an active repair process of the intestinal barrier disrupted by the parasite. The gross pathology observed at 21 dpe, where the intestine took on a rigid, leathery appearance and was mechanically resistant to homogenization in liquid nitrogen might be related to the thickening of the lamina propria-submucosa due to the observed hyperplasia. The increased vascularization of the organ that we observed would contribute to the tissue repair process. By 21 dpe, no parasite stages were observed beyond the stratum compactum, which appeared to act as a parasite barrier. Thus, parasite proliferation is limited in resistant steelhead trout to the mucosal layers of the intestine, where cell regeneration occurs on a daily basis favoring lesion recovery. This is likely aided by the massive upregulation of IL-10 at 21 dpe, which would facilitate the resolution of inflammation and subsequent tissue repair. This is similar to what is observed in gilthead sea bream that become resistant to reinfection with *E. leei*, where IL-10 is associated with the resolution of infection [45]. The tissue response of resistant steelhead trout in this study strongly contrasts with what is observed in susceptible fish, where all layers of the intestine become infected and it becomes a soft, spongy mass that loses its overall structure [28,46,47]. How the stratum compactum becomes a physical barrier and whether it contributes to the intestinal hardening, deserves further studies. It does appear that the ability of resistant fish to maintain their intestinal structure in the face of parasite replication is likely a critical factor in resisting *C. shasta* induced mortality and would explain why previous studies have observed organized, tissue-level responses to *C. shasta* in resistant fish, but not in susceptible fish [48,49,50,51].

Differences in the intestinal epithelial integrity between hosts have also been observed in studies of the intestinal myxozoans *Enteromyxum scophthalmi* and *E. leei* [52]. Turbot is highly susceptible to *E. scophthalmi*, suffering serious intestinal lesions and barrier dysfunction, leading to high morbidity and mortality rates. Gilthead sea bream (*Sparus aurata* L.), on the other hand, experience low mortality rates and are able to better maintain their intestinal epithelial integrity even when heavily infected by *E. leei* [53]. Similar to *C. shasta*, *E. leei* is able to infect a wide range of fish species with varying degrees of host susceptibility [4]. For sharpsnout seabream (*Diplodus puntazzo*), differences in humoral immune factors have been suggested to play a role in their high susceptibility to *E. leei* [54], however no transcriptomic comparison with gilthead sea bream has been conducted. It should also be noted that gilthead sea bream that survive primary exposure to *E. leei* acquire a protective immunity to reinfection, that is associated with increased IgM and IgT expression relative to naïve fish [45]. Similar to gilthead sea bream, the resistant fish in this study were better able to maintain their intestinal integrity and had higher expression of IgM and IgT compared to susceptible fish at this same point in the infection. 

Given how different the intestinal response of resistant fish is at 7 dpe compared to susceptible fish, it is surprising how similar the response is at 21 dpe. 7820 genes were differentially expressed in both phenotypes and their expression levels were highly correlated. In addition to numerous metabolic and cell junction genes, this shared response includes several immune factors (IL-6, IL-8, IL-10, IFN-γ) that have been found to be upregulated in other studies of *C. shasta* infected salmonids [24,25,55]. While we cannot rule out temporal differences in the expression of these genes being a critical factor in resistance, something that almost certainly occurs given the delayed parasite recognition of susceptible fish, their differential expression alone does not explain resistance vs. susceptibility to *C. shasta*. Examination of the genes that are only differentially expressed in resistant fish at 21 dpe revealed a much larger role for the adaptive immune response in these fish, particularly the B cell response. Resistant fish have significantly more heavy and light chain transcripts upregulated, as well as secreted IgM and IgT transcripts at this time. Again, this is likely influenced by the delayed parasite recognition of susceptible fish, but it suggests an earlier and stronger antibody response at play in resistant fish. It has been demonstrated that salmonids are capable of generating IgM and IgT that is specific to *C. shasta* [56]. However, the effectiveness of this antibody response in reducing mortality from *C. shasta* remains unclear. A study of susceptible rainbow trout exposed to *C. shasta* showed upregulation of IgM and IgT in these fish, but it failed to improve their condition and 100% mortality occurred [55]. The authors noted that this may be due to the response coming too late, after the intestine has been severely damaged. Our observations would support this, as resistant fish were better able to maintain their intestinal structure and mounted an earlier adaptive immune response, including the observed foci of lymphocyte-like cells. 

Differences in the T cell response are also evident at 21 dpe, most notably among T_H_17 cytokines which are an important aspect of the gut mucosal barrier that help prevent the dissemination of bacteria [57]. In our previous study, we observed a strong downregulation of IL-17 family cytokines in susceptible fish at 21 dpe, whereas in the resistant fish in this study, no IL-17 cytokines were downregulated and IL-17A was upregulated. Whether this is directly related to the *C. shasta* is unclear and it may be more related to overall gut health and the invasion of opportunistic bacteria. More interesting is the possible transition to a T_H_2 response that may be occurring late in the infection. This adaptive immune response is driven by IL4/13, which we observed massive upregulation of at 21 dpe, along with upregulation of the transcriptional regulators GATA3 and STAT5. The T_H_2 response is associated with wound healing, the suppression of T_H_1-driven inflammation, and the elimination of helminth parasites [58,59]. Although much smaller than helminth worms, *C. shasta* is also an extracellular intestinal parasite and the mechanisms used to clear helminths (B cell activation, eosinophil recruitment, mucus hypersecretion, increased cellular turnover) may help eliminate or otherwise create an unfavorable environment for *C. shasta*.

Comparison of our finding with other studies on myxozoan infections suggest that an effective T cell response is critical for reducing pathology. Similar to what we observed in *C. shasta* infected rainbow trout, when strains of rainbow trout that are resistant and susceptible to whirling disease are compared, the resistant strain has an earlier and more effective immune response, with upregulation of IFN-gamma sooner in the infection and a stronger T cell response during the initial stages [26,60]. The failure of this early immune response in susceptible fish leads to a sustained inflammatory response that fails to control the infection and likely contributes to tissue damage. The pathology of proliferative kidney disease in rainbow trout is associated with an early imbalance of T_H_ cytokines and a dysregulated B cell response [9]. An in-depth analysis of T_H_ cytokines in gilthead sea bream during *E. leei* infection revealed a strong induction of T_H_1 and T_H_17 cytokines in the intestine [61]. The ability of Atlantic salmon (*Salmo salar*) to resolve infection with *K. thyrsites*, and develop resistance to reinfection, is linked to a cytotoxic T cell response [7]. Further examination of the T cell response during *C. shasta* infection, including determination of the cellular kinetics and the specific T cell subsets responding to the infection, would help elucidate the cellular basis to resistance.

## 4. Conclusions

In this study we employed RNA-seq to explore the complex dynamic between host and parasite and to understand how resistant steelhead trout are able to overcome an actively progressing *C. shasta* infection in their intestine. Our results confirmed that early parasite recognition is critical for resistance and that initial invasion of the intestine by the parasite elicits a strong IFN-gamma driven adaptive immune response in resistant fish. This is accompanied by remodeling of the host intestinal tissue, which is likely vital for maintaining the overall intestinal structure in the face of extensive parasite replication and the tissue inflammation that results. This early adaptive immune response leads to a vigorous B cell response later in the infection, characterized by strong antibody production, particularly of IgT. Based on these results, we propose that a core immune response to *C. shasta* exists among resistant and susceptible fish and that temporal differences in the expression of key immune factors (IFN-γ, IL-6, GIMAP4, IgT), resulting from earlier parasite recognition by resistant fish, largely explains the different infection outcomes for these fish. In conjunction with this, resistant fish have a different response to the parasite at the tissue level with the stratum compactum playing an important role in limiting parasite spreading, which might be related to the induction of keratinization. Given that most RNA-seq studies have been conducted on fish that are susceptible to myxozoans, we believe the present study offers a valuable framework for putting those, and future studies, in perspective.

## 5. Methods

### 5.1. Experimental Fish

Resistant steelhead trout were collected from the Round Butte Hatchery (OR, USA) and bred as previously described [62]. The fish were reared on 13.5° C specific-pathogen free (SPF) well water at the Oregon State University (OSU) John L. Fryer Aquatic Animal Health Laboratory (AAHL) in Corvallis, OR, USA, and fed a commercial diet daily (BioClark’s Starter, Bio-Oregon, Longview, WA, USA).

### 5.2. Parasite Challenge

Laboratory cultures of the invertebrate host *Manayunkia occidentalis* are maintained at the AAHL and serve as source of *C. shasta* actinospores for laboratory challenges. The annelids are housed in indoor mesocosms receiving flow-through UV-treated treated river water and spore production is routinely monitored [63]. The parasite challenge was initiated in March 2019, when spore production from the genotype IIC mesocosm reached 30,000 spores per day, a sufficient dose to meet or exceed their threshold of resistance [11,20]. 21 resistant steelhead trout (average 103.1 g ± 7.1 g) were placed in a 100-liter tank that received effluent from the mesocosm. Water temperature was increased from 14.5 ℃ to 17.5 ℃ ± 0.4 ℃ over 2 days by addition of 18 ℃ SPF well-water. This temperature was chosen as it is representative of the river water temperatures that out-migrating salmonids experience when they are exposed to *C. shasta*, and it aligns with our previous study of the transcriptomic response of steelhead trout. An additional 21 Round Butte steelhead trout were transferred into an identical tank setup but received effluent from the control mesocosm which contains uninfected *M. occidentalis*. 5 days after the exposure began, the fish (treatment and control) were transferred into six 25-liter tanks (7 fish per tank) that were randomly assigned and supplied with 18 ℃ SPF well-water.

### 5.3. Tissue Sampling

Fish were sampled at 7- and 21-days post exposure (dpe), with exposure being defined as their initial placement in the exposure tank. At each timepoint, 3 fish from each of the 6 tanks were euthanized with an overdose of MS-222 (tricaine methanosulfonate, Argent Laboratories, Redmond, WA, USA) for a total of 18 fish per timepoint (9 exposed, 9 controls). The entire intestine was removed from each fish and placed in either RNAlater (2 out 3 fish per tank) or Dietrich’s fixative (1 out of 3 fish per tank). The samples collected in RNAlater were immediately stored at 4 ℃ and then transferred to –80 ℃ after 24 hours. At each timepoint, the fish were sampled at the same time of day to avoid changes in gene expression due to circadian rhythms [64]. After the fish were sampled at the 21 dpe timepoint, the remaining fish in each tank were consolidated into two tanks (exposed and control) to eliminate distress due to isolation. The fish were monitored until 60 dpe, at which time they were euthanized with an overdose of MS-222.

### 5.4. Sample Processing

The intestine samples collected in RNAlater were homogenized in liquid nitrogen using a porcelain mortar and pestle. Following this, 25 mg of homogenized tissue from each sample underwent RNA extraction using the RNeasy Mini Kit (Qiagen, catalog number 74104) according to the manufacture’s protocol. An additional 25 mg of tissue underwent DNA extraction using the DNeasy Blood and Tissue Kit (Qiagen, catalog number 69506). The extracted DNA was eluted in 30 μl of Buffer AE that was applied to the spin column twice to achieve a higher concentration. The concentration and purity and of the extracted RNA and DNA was assessed with a NanoDrop ND-1000 UV-Vis Spectrophotometer. 

The amount of parasite DNA present in the intestine was determined using *C. shasta*-specific qPCR assay [63] and 100 ng of extracted DNA from each sample was assayed in triplicate wells through 40 cycles using an Applied Biosystems StepOnePlus Real-Time PCR System. A sample was considered positive for *C. shasta* if all three wells fluoresced, and the sample was rerun if the Cq standard deviation between wells was greater than 1. On each qPCR plate, a positive control, a negative control (molecular grade water), and a standard dilution curve equivalent to 1, 10, 100, and 1000 actinospores was included.

Intestinal samples in Dietrich’s fixative were routinely processed for histology, embedded in Technovit 7100 resin (Heraeus Kulzer, Wehrheim, Germany) and 3 µm sections were stained with Giemsa or with periodic acid Schiff (PAS). Observations and microphotographs were made with a Leitz Dialux 22 light microscope connected to an Olympus DP70 camera. 

### 5.5. Library Prep and Sequencing

Intestinal mRNA from 8 samples at each timepoint (4 treatment, 4 control) were submitted to the Center for Genome Research and Biocomputing at OSU for library preparation and sequencing. The integrity of the RNA was confirmed by running each sample on an Agilent Bioanalyzer 2100 (Agilent Technologies, Santa Clara, CA, USA). A totol of 1 μg of RNA was used for library preparation using the Illumina TruSeq™ Stranded mRNA LT Sample PrepKit according to the manufacturer’s instructions (Cat. No. RS-122-2101, Illumina Inc. San Diego, CA, USA). Library quality was checked with a 4200 TapeStation System (Agilent Technologies, USA) and quantified via qPCR. The libraries were sequenced on two lanes of an Illumina HiSeq 3000 as 100-bp single-end runs.

### 5.6. Data Analysis

Adapter sequences were trimmed from the raw reads using BBDuk (25 January 2018 release) [65], and reads less than 30-bp after trimming were discarded. Library quality was assessed before and after trimming using FastQC (v 0.11.8, https://www.bioinformatics.babraham.ac.uk/projects/fastqc/, accessed on 30 June 2020) [66]. The trimmed reads were then mapped to the rainbow trout reference genome (GenBank: MSJN00000000.1) using HiSat2 (v 2.1.0, http://daehwankimlab.github.io/hisat2/, accessed on 26 July 2020) [67]. HTSeq-count (v 0.11.1, https://htseq.readthedocs.io/en/release_0.11.1/index.html, accessed on 26 July 2020) [68] was used to calculate the number of reads that mapped to each gene and the counts were imported into R 3.5.0 [69] and loaded into the package DESeq2 (v 1.22.2, https://bioconductor.org/packages/release/bioc/html/DESeq2.html, accessed on 26 July 2020) [70]. To assess the similarity of the sequenced samples, a PCA plot was constructed using normalized gene counts that were transformed using the rlog function in DESeq2. Hierarchical clustering was performed with the Pheatmap package.

Differentially expressed genes (DEGs) between treatment and control fish were identified using the negative binomial Wald test in DESeq2 and were considered significant if they had a Benjamini–Hochberg false discovery rate (FDR) adjusted *p* value < 0.05 and an absolute log_2_(fold change) > 1. Annotation of the DEGs and gene ontology (GO) enrichment was conducted with OmicsBox (v 1.1.135, https://www.biobam.com/omicsbox/, accessed on 12 August 2020) [71]. To obtain high quality annotations, the blast e-value cutoff was set at 1 × 10^−5^ and genes were preferentially annotated with the SWISS-PROT database [72] followed by the NCBI nonredundant database and the ‘Vertebrata’ taxonomy filter was applied. All genes detected in this study were used as the background for GO enrichment and up- and downregulated genes were analyzed separately. Enriched GO terms and their FDR-adjusted *p*-values were imported into Cytoscope (v 3.7.2, https://cytoscape.org/, accessed on 20 August 2020) [73] for visualization with the ClueGo (v 2.5.6, http://www.ici.upmc.fr/cluego/, accessed on 20 August 2020) [74] plugin, which clusters the GO terms into functionally related networks. *O. mykiss* was chosen as the organism for Ontologies/Pathways and the GO Term Fusion option was used to merge GO terms based on similar associated genes. Volcano plots were constructed with the R package EnhancedVolcano (v 1.0.1, https://bioconductor.org/packages/release/bioc/html/EnhancedVolcano.html, accessed on 12 September 2020) [75].

DEGs found in this study were compared to DEGs previously identified in susceptible steelhead trout exposed to *C. shasta* [62]. Genes that were differentially expressed in both phenotypes at 21 dpe were identified and the Pearson correlation coefficient between their expression levels was calculated in R. To analyze the antibody response against *C. shasta*, sequenced reads from both resistant and susceptible at 21 dpe were mapped against the coding sequence for the secreted forms of IgM (GenBank: S63348.1) and IgT (GenBank: AY870263.1) using Salmon (v 0.10.0, https://salmon.readthedocs.io/en/latest/, accessed on 21 September 2020) [76]. The output was imported into DESeq2 to estimate fold-change between exposed and control fish.

## Figures and Tables

**Figure 1 pathogens-10-01179-f001:**
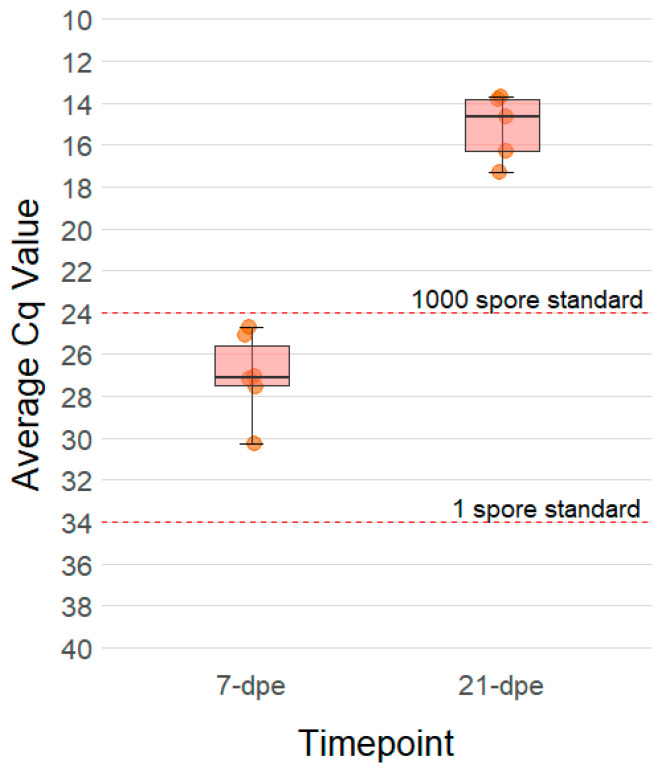
Boxplot showing the relative quantity of *Ceratonova shasta* DNA present in the intestines of resistant steelhead trout at 7- and 21-days post exposure (dpe). Each dot represents the average quantification cycle (Cq) of 100 ng of DNA extracted from the intestines of one fish that was assayed in triplicate by qPCR. Six fish were assayed at each timepoint. Dashed red lines indicate the average Cq values obtained from 1 and 1000 actinospore standards.

**Figure 2 pathogens-10-01179-f002:**
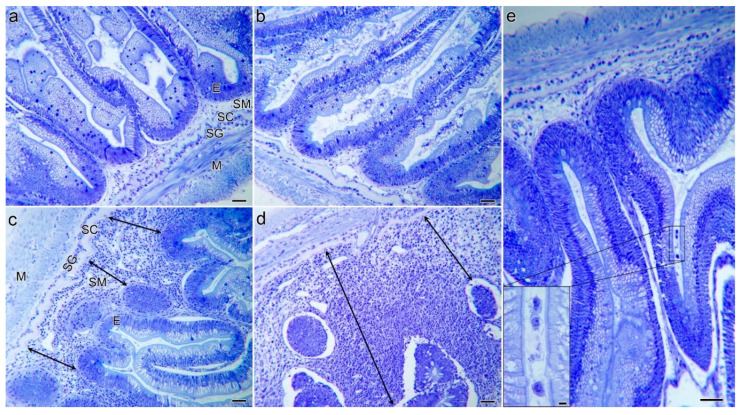
Intestinal structure of unexposed control (**a**,**b**) and infected (**c**–**e**) resistant steelhead trout at 7 dpe to *Ceratonova shasta* (**a**,**c**), at 21 days (**b**,**d**) and 60 dpe (**e**). Note the mild (**c**) and severe (**d**) submucosal hyperplasia in recipient fish at the two early time points (double arrowheads). At 60 dpe, the intestinal tissue structure has recovered and resembles that of controls (**e**). Note the presence of parasite stages in the intestinal lumen (insert). Giemsa staining. Scale bars = 50 µm, except in insert = 5 µm. E; epithelium, SM; intestinal submucosa; SC; stratum compactum, SG; stratum granulosum, M; muscularis.

**Figure 3 pathogens-10-01179-f003:**
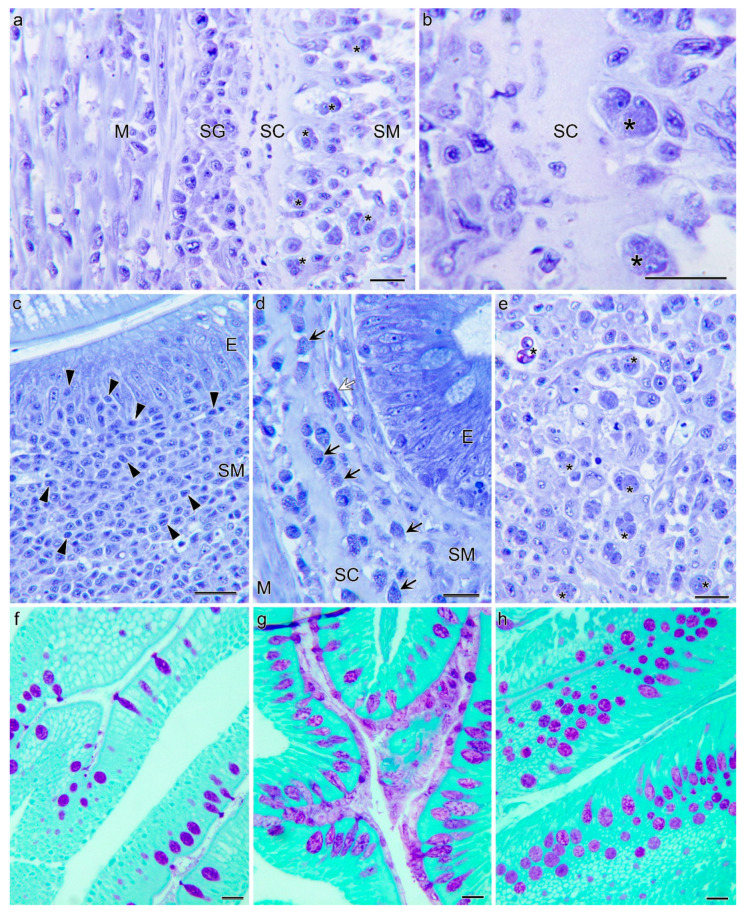
Intestinal histopathology of resistant steelhead trout exposed to *Ceratonova shasta*. At 21 days post exposure (dpe) (**a**,**b**,**e**) parasite stages (asterisks) present in the intestinal submucosa (SM) reach the stratum compactum (SC) but do not invade the underlying stratum granulosum (SG) and muscularis (M). Note that the lamina propria-submucosa is invaded by the parasite and has lost its tissue structure (**e**). Inflammatory cell proliferation and infiltration at 7 dpe consisting of lymphocyte-like cells (**c**). Cluster of granulocytes (black arrows) including one evident eosinophil (white arrow) in the submucosa at a 60 dpe (**d**). Mucins stained with periodic acid Schiff (PAS) (magenta) in 7 dpe (**f**), 21 dpe (**g**) and 60 dpe (**h**) intestines. Note the high abundance of PAS+ goblet cells at the two latter time points. Giemsa staining (a–e). Scale bars = 20 µm. E; epithelium.

**Figure 4 pathogens-10-01179-f004:**
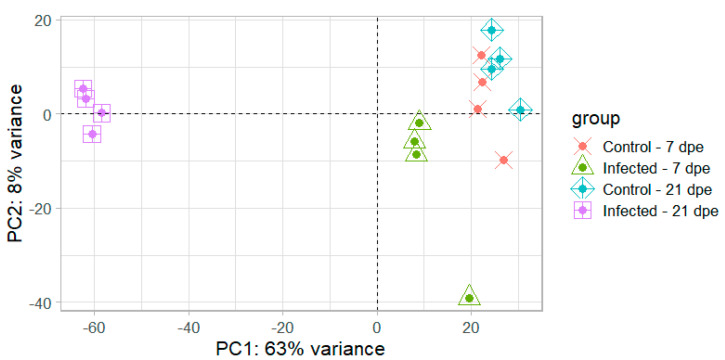
Principal component analysis of the regularized log transformed gene counts for each sample. dpe = days post exposure.

**Figure 5 pathogens-10-01179-f005:**
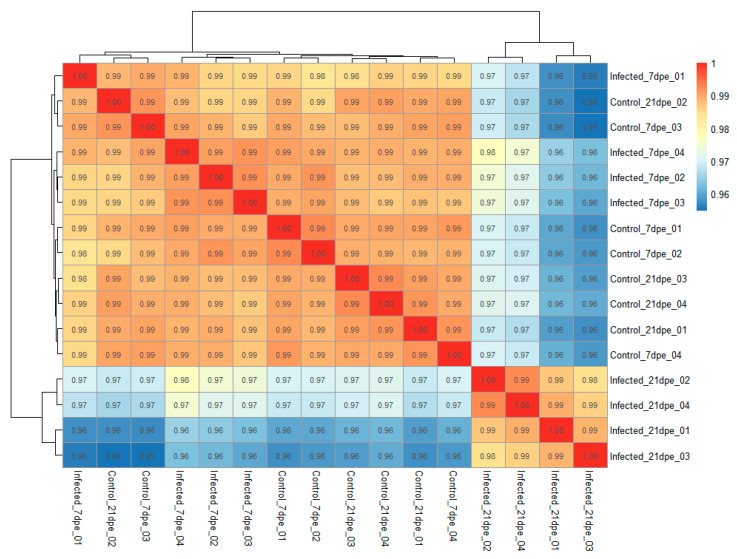
Hierarchical clustering and heatmap showing the Pearson correlation coefficient (r) between each sample. Values were generated from rlog transformed gene counts for each sample using the R packages DESeq2 and Pheatmap.

**Figure 6 pathogens-10-01179-f006:**
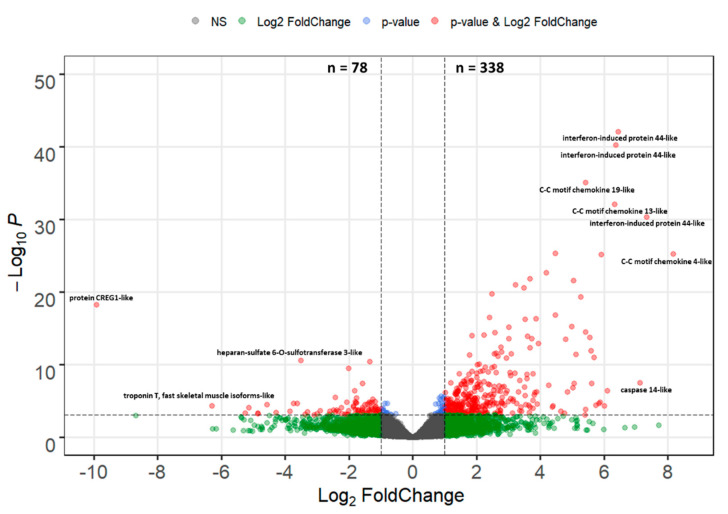
Volcano plot of the differential gene expression in the intestine of resistant steelhead trout at 7 days post exposure to *Ceratonova shasta*. Each dot represents the average value of one gene across four biological replicates. Red indicates the gene was significant at the FDR-adjusted *p* value and Log2-Foldchange threshold, blue is significantly only by *p* value, green only by Log2-FoldChange, and gray were not significant by either metric.

**Figure 7 pathogens-10-01179-f007:**
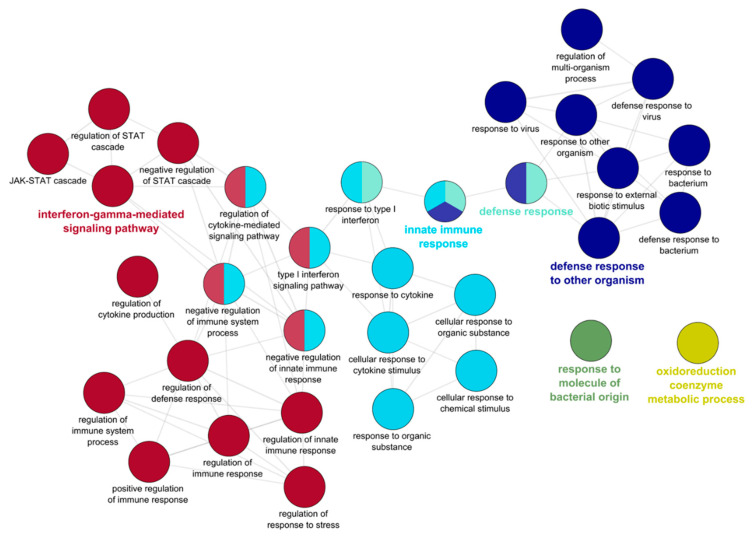
Gene ontology (GO) enrichment of the genes upregulated in the intestine of resistant steelhead trout at 7 days post exposure to *Ceratonova shasta*. Enriched GO terms were grouped into functionally related nodes using ClueGO, a Cytoscope plugin. Nodes are colored and grouped according to related functions and labelled by the most significant term of the group. Node size corresponds to the FDR-adjusted *p* value of each GO term and is specific to each graph.

**Figure 8 pathogens-10-01179-f008:**
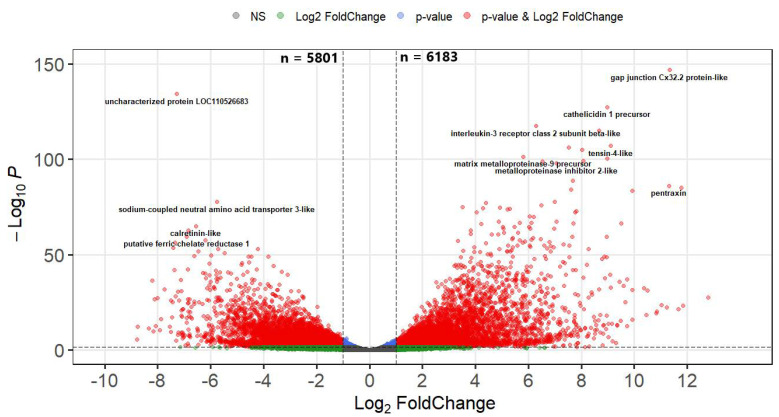
Volcano plot of the differential gene expression in the intestine of resistant steelhead trout at 21 days post exposure to *Ceratonova shasta*. Each dot represents the average value of one gene across four biological replicates. Red indicates the gene was significant at the FDR-adjusted *p* value and Log2-Foldchange threshold, blue is significantly only by *p* value, green only by Log2-FoldChange, and gray were not significant by either metric.

**Figure 9 pathogens-10-01179-f009:**
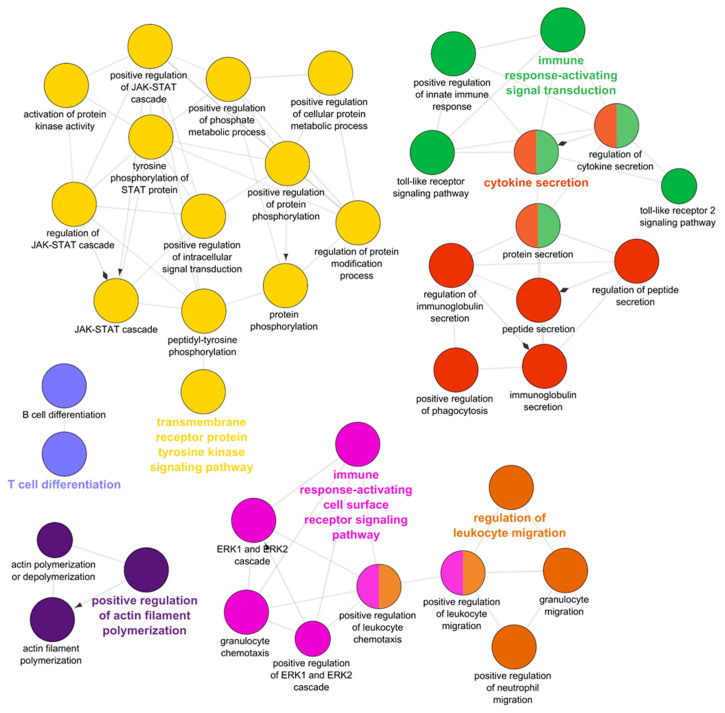
Gene ontology (GO) enrichment of the genes upregulated in the intestine of resistant steelhead trout at 21 days post exposure to *Ceratonova shasta*. Enriched GO terms were grouped into functionally related nodes using ClueGO, a Cytoscope plugin. Nodes are colored and grouped according to related functions and labelled by the most significant term of the group. Node size corresponds to the FDR-adjusted *p*-value of each GO term and is specific to each graph.

**Figure 10 pathogens-10-01179-f010:**
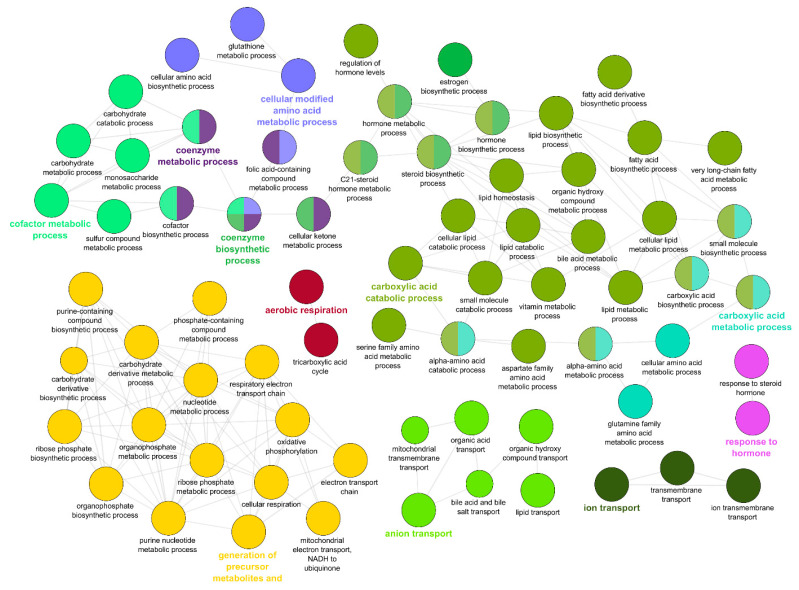
Gene ontology (GO) enrichment of the genes downregulated in the intestine of resistant steelhead trout at 21 days post exposure to *Ceratonova shasta*. Enriched GO terms were grouped into functionally related nodes using ClueGO, a Cytoscope plugin. Nodes are colored and grouped according to related functions and labelled by the most significant term of the group. Node size corresponds to the FDR-adjusted *p*-value of each GO term and is specific to each graph.

**Figure 11 pathogens-10-01179-f011:**
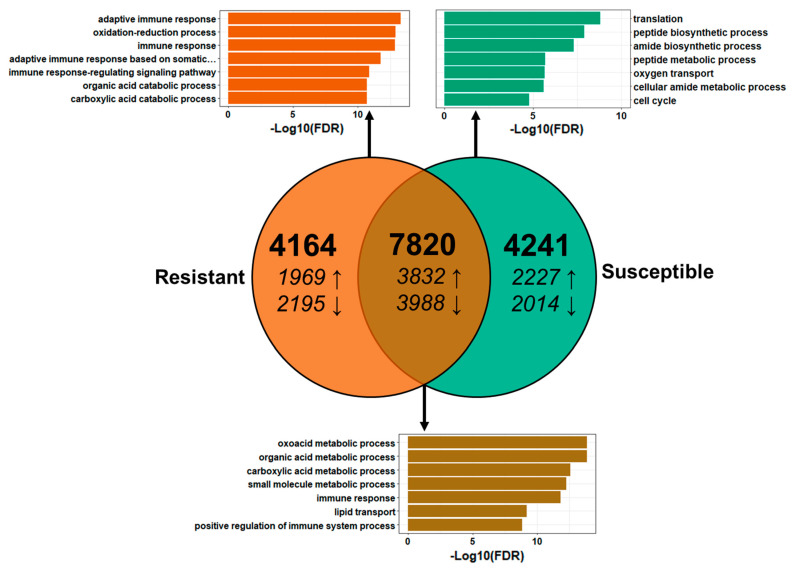
Comparison of the differential gene expression between resistant and susceptible steelhead trout at 21 days post exposure to *Ceratonova shasta*. Venn diagram showing the number of differentially expressed genes (DEGs) shared between resistant and susceptible steelhead trout. Arrows indicate upregulation vs. downregulation. The top seven most statistically significant enriched GO terms for each set of DEGs are shown in the corresponding bar graphs.

**Table 1 pathogens-10-01179-t001:** Average number of sequencing reads generated per sample for each group and the corresponding number of reads that could be mapped to one locus in the rainbow trout genome.

Group	Average Number of Reads	Average Number of Mapped Reads
Controls—7 dpe	29,536,649	21,403,620 (72.5%)
Infected—7 dpe	24,588,191	17,768,446 (72.2%)
Controls—21 dpe	44,080,197	33,526,343 (75.4%)
Infected—21 dpe	31,730,221	11,964,621 (40.4%)

**Table 2 pathogens-10-01179-t002:** Selected genes that were differentially expressed in the intestine of resistant steelhead trout at 7 dpe to *Ceratonova shasta*.

Entrez Gene ID	Protein Product	Fold Change
Cell Adhesion and Migration		
LOC110493534	beta-parvin-like	2.2
LOC110530422	cadherin-2-like	2.5
LOC110488853	CD97 antigen-like	6.3
LOC110537995	cell adhesion molecule 4-like	3.7
LOC110534540	claudin-1-like	4.0
LOC110524880	E-selectin-like	3.8
LOC110531988	fibulin-2-like	6.4
LOC110529577	gap junction C × 32.2 protein-like	2.7
LOC110495444	gap junction gamma-1 protein-like	−4.1
LOC110537004	integrin alpha-2-like	3.6
LOC100653476	integrin alpha-4	2.9
LOC110528502	integrin alpha-5-like	3.8
LOC110509064	peroxidasin-like protein	13.5
LOC110486368	vascular cell adhesion protein 1-like	2.9
Antigen Processing and Presentation		
LOC110488346	proteasome subunit beta type-7-like	2.1
LOC110488347	proteasome subunit beta type-8-like	2.5
LOC110488348	proteasome subunit beta type-9	2.6
LOC110528193	E3 ubiquitin-protein ligase DTX3L-like	2.2
LOC110523717	E3 ubiquitin-protein ligase NEURL3-like	31.7
LOC110487518	E3 ubiquitin-protein ligase RNF135-like isoform X3	14.1
LOC110537403	E3 ubiquitin-protein ligase RNF144A-like	2.6
LOC110485543	E3 ubiquitin-protein ligase RNF144B-like	2.8
LOC110524942	E3 ubiquitin-protein ligase rnf213-alpha-like	5.9
LOC110531504	E3 ubiquitin-protein ligase TRIM39-like	9.3
LOC110497030	E3 ubiquitin-protein ligase TRIM69-like	12.5
LOC110534395	probable E3 ubiquitin-protein ligase HERC3	7.9
LOC110500498	probable E3 ubiquitin-protein ligase RNF144A-A	5.2
LOC110500501	ubiquitin carboxyl-terminal hydrolase 47-like	3.1
LOC110536452	ubiquitin-conjugating enzyme E2 L3-like	2.1
LOC110530687	ubiquitin-conjugating enzyme E2 R1-like	2.0
LOC110494671	ubiquitin-like modifier-activating enzyme 1	3.4
LOC100136299	tapasin long form precursor	5.8
LOC100136300	tapasin-related	3.4
LOC110500638	tapasin-related protein-like	4.0
tap2a	TAP2a protein	2.1
LOC110535123	endoplasmic reticulum aminopeptidase 1-like	3.4
LOC110526886	endoplasmic reticulum aminopeptidase 2-like	3.0
LOC110487866	antigen peptide transporter 1-like	3.2
LOC110488345	antigen peptide transporter 2-like	6.1
ly75	lymphocyte antigen 75 precursor	3.2
LOC110504514	lymphocyte function-associated antigen 3-like	2.0
LOC110514021	CD83 antigen-like	8.3
LOC110501369	major histocompatibility complex class I-related gene protein-like	3.4
LOC110506874	major histocompatibility complex class I-related gene protein-like	5.8
Chemokines		
LOC110512719	C-C chemokine receptor type 5-like	5.0
LOC110530382	C-C chemokine receptor type 8-like	2.7
LOC100135979	CC chemokine with stalk CK2 precursor	4.9
LOC110514657	C-C motif chemokine 13-like	79.9
LOC110536449	C-C motif chemokine 19-like	51.4
LOC110536450	C-C motif chemokine 4-like	285.6
cxcf1a	chemokine CXCF1a precursor	4.7
cxcf1b	chemokine CXCF1b precursor	3.8
cxcl13	chemokine CXCL13 precursor	13.6
LOC100136107	chemokine receptor-like protein 1	4.2
LOC110509178	chemokine XC receptor 1-like	2.5
LOC110485791	C-X-C motif chemokine 11-like	12.7
ccl13	Small inducible cytokine A13 precursor	33.0
Interferon Stimulated Genes		
irf-1	interferon regulatory factor 1	11.4
LOC110533376	interferon regulatory factor 1-like	2.3
LOC110492403	interferon regulatory factor 4-like	5.6
LOC110526480	interferon regulatory factor 8-like	3.5
LOC110538600	interferon-induced 35 kDa protein homolog	2.3
mx	interferon-induced GTP-binding protein Mx1	5.2
LOC110520643	interferon-induced GTP-binding protein Mx-like	13.9
LOC110494016	interferon-induced protein 44-like	161.2
LOC110487764	stimulator of interferon genes protein-like	4.9
Pattern Recognition Receptors		
LOC110489499	C-type lectin domain family 4 member E-like	2.7
LOC110517676	C-type lectin domain family 9 member A-like	11.5
LOC110508265	macrophage mannose receptor 1-like	12.8
LOC110496441	toll-like receptor 13	12.6
LOC110507636	high affinity immunoglobulin gamma Fc receptor I-like	3.3
LOC110523269	protein NLRC5	3.9
LOC110531495	fucolectin-4	2.3
Cytokines		
socs1	suppressor of cytokine signaling 1	4.0
LOC110495002	suppressor of cytokine signaling 1-like	12.0
LOC110512513	suppressor of cytokine signaling 3-like	2.9
LOC110520020	signal transducer and activator of transcription 1-alpha/beta-like	2.4
ifngamma2	interferon gamma 2 precursor	32.4
LOC100135968	interleukin 13 receptor alpha-2	5.9
il12b	interleukin-12 beta chain precursor	5.3
LOC110500135	interleukin-15 receptor subunit alpha-like	2.2
il2rc2	interleukin-2 receptor gamma chain-2 precursor	3.7
Adaptive immune response		
LOC101268951	tumor necrosis factor receptor superfamily member 5	4.0
tnfrsf5a	TNF receptor superfamily member 5A precursor	2.7
LOC110497707	TNF receptor-associated factor 3-like	2.9
LOC110496128	BLIMP-1	3.6
LOC110537828	B-cell antigen receptor complex-associated protein alpha chain-like	4.4
LOC110537869	B-cell lymphoma 3 protein homolog	2.6
LOC110516594	B-cell receptor CD22-like	6.7
LOC110498810	leukocyte antigen CD37-like	4.0
LOC110485442	GTPase IMAP family member 4-like	3.8
LOC110533001	GTPase IMAP family member 7-like	12.0
Proteases and Protease Inhibitors		
LOC110521419	mast cell protease 2-like	8.4
LOC110530599	matrix metalloproteinase-14-like	3.4
LOC110494201	matrix metalloproteinase-19-like	3.9
LOC110491324	matrix metalloproteinase-25-like	6.7
LOC110538774	leukocyte elastase inhibitor-like	2.1
LOC110538848	metalloproteinase inhibitor 2-like	2.2
Apoptosis		
LOC110536971	programmed cell death 1 ligand 1-like	3.8
LOC110495001	programmed cell death protein 4-like	48.8
LOC110537938	apoptosis regulator BAX-like	2.3
LOC110491862	tumor necrosis factor receptor superfamily member 6B-like	8.4
LOC110503799	caspase-1-like	2.9
LOC100135902	caspase 6 precursor	3.2
LOC110523645	caspase recruitment domain-containing protein 8-like	4.8
LOC110492801	calpain-1 catalytic subunit-like	2.4
LOC110534030	calpain-5-like	5.7
Others		
LOC110511404	caspase-14-like	139.2
LOC110511405	caspase-14-like	34.7
LOC110506002	arginase-2, mitochondrial-like	4.8
e7	type I keratin E7	5.0
LOC100136204	cathelicidin 1 precursor	38.1
LOC110523368	pyrin-like	6.8
LOC100135935	hepcidin	28.6

**Table 3 pathogens-10-01179-t003:** Selected immune genes that were differentially expressed in the intestine of resistant steelhead trout at 21 days post exposure to *Ceratonova shasta*.

Entrez Gene ID	Protein Product	Fold Change
T-cells		
cd3z	T-cell surface glycoprotein CD3 zeta chain	8.0
LOC100136285	T-cell surface glycoprotein CD4	5.6
onmy-cd8a	T-cell surface glycoprotein CD8 alpha precursor	−9.2
LOC100136222	CD8 beta precursor	−6.1
LOC110495954	T-cell-specific surface glycoprotein CD28-like	9.8
LOC110500642	CD276 antigen-like	217.3
LOC100136275	CTLA4-like protein precursor	44.7
LOC110509520	lymphocyte activation gene 3 protein-like	377.5
prkcq	protein kinase C theta type	12.5
LOC110536971	programmed cell death 1 ligand 1-like	65.4
LOC110489621	V-set domain-containing T-cell activation inhibitor 1-like	−10.1
LOC110516762	GTPase IMAP family member 7-like	38.9
T_H_1		
ifng	interferon gamma precursor	90.9
ifngamma2	interferon gamma 2 precursor	249.0
ifngr1	interferon gamma receptor 1	16.1
ifngr1	interferon-gamma receptor alpha chain precursor	3.7
irf-1	interferon regulatory factor 1	6.2
LOC110506608	interferon regulatory factor 8-like	8.6
il12b	interleukin-12 beta chain precursor	89.2
LOC110524480	interleukin-12 receptor subunit beta-2-like	−3.1
LOC110524481	interleukin-12 receptor subunit beta-2-like	9.8
LOC110508876	interleukin-12 subunit alpha-like	47.2
LOC110489482	interleukin-12 subunit beta-like	−4.2
LOC110511354	interleukin-18 receptor accessory protein-like	20.2
LOC110501544	signal transducer and activator of transcription 1-alpha/beta-like	6.4
tbx21	T-box 21	33.0
T_H_2		
il4/13a	interleukin-4/13A precursor	52.9
LOC110489171	interleukin-4/13b1 precursor	72.8
LOC110504551	interleukin-4/13b2 precursor	774.4
LOC110500122	transcription factor GATA-3-like	6.1
stat5	signal transducer and activator of transcription 5	7.7
socs3	suppressor of cytokine signaling 3	46.4
T_H_17		
LOC110529296	interleukin-17A-like	25.6
LOC110524663	interferon regulatory factor 4-like	31.5
LOC110520784	nuclear receptor ROR-gamma-like	−22.6
LOC110535950	nuclear receptor ROR-gamma-like	−16.0
T-reg		
il10	interleukin-10 precursor	1393.7
LOC100136774	transforming growth factor beta-1	3.9
Cytotoxic and NK cells		
LOC110538116	perforin-1-like	106.3
LOC110536463	granzyme A-like	10.3
LOC110520655	granzyme B-like	7.7
LOC110524258	granzyme-like protein 2	7.2
LOC110500840	natural killer cell receptor 2B4-like	95.1
LOC110498133	antimicrobial peptide NK-lysin-like	31.8
B-cells		
LOC110522002	BLIMP-1	223.7
LOC110496128	BLIMP-1	55.1
LOC110499048	tumor necrosis factor receptor superfamily member 13B-like	1821.4
LOC110491449	tumor necrosis factor receptor superfamily member 13B-like	7.8
LOC110494997	B-cell receptor CD22-like	37.7
LOC110537828	B-cell antigen receptor complex-associated protein alpha chain-like	24.3
LOC110537869	B-cell lymphoma 3 protein homolog	19.5
LOC110521598	polymeric immunoglobulin receptor-like	69.7
nilt2	polymeric immunoglobulin receptor-like precursor	-3.4
cd79b	CD79b	5.4
klhl6	kelch-like protein 6	16.5
Macrophages		
LOC110520098	interleukin-34-like	6.2
csf1	macrophage colony-stimulating factor precursor	4.0
LOC110508265	macrophage mannose receptor 1-like	978.1
LOC110508267	macrophage mannose receptor 1-like	224.3
LOC110536912	macrophage mannose receptor 1-like	−159.8
LOC100136664	macrophage myristoylated alanine-rich C kinase-like protein	34.6
LOC110520391	macrophage receptor MARCO-like	4.3
LOC100136179	arginase-1	36.5
LOC110498289	arginase-2, mitochondrial-like	42.6
**Granulocytes**		
csf-3	granulocyte colony-stimulating factor precursor	143.1
LOC100136240	neutrophil cytosolic factor 2	21.7
LOC110523686	eosinophil peroxidase-like	108.4
LOC110524274	mast cell protease 1A-like	14.6
Cytokines		
LOC110536401	interleukin-1 beta-like	965.1
il-6	interleukin-6 precursor	1851.8
LOC110496949	interleukin-6-like	7118.8
il-8	putative CXCL8/interleukin-8	476.2
socs1	suppressor of cytokine signaling 1	9.6
LOC110532426	suppressor of cytokine signaling 2-like	−7.2
socs5	suppressor of cytokine signaling 5	2.3
tnf	tumor necrosis factor	12.2
Chemokines		
cxcl13	chemokine CXCL13 precursor	3258.9
LOC110490829	C-C chemokine receptor type 3-like	1443.9
LOC110494122	C-C motif chemokine 4 homolog	1313.9
LOC100135979	CC chemokine with stalk CK2 precursor	403.6
LOC110514657	C-C motif chemokine 13-like	232.9
LOC110485791	C-X-C motif chemokine 11-like	173.5
LOC110525193	C-C motif chemokine 19-like	83.1
Other		
LOC100653444	hepcidin-like	2399.5
LOC110490701	complement C1q tumor necrosis factor-related protein 3-like	2098.0
LOC100136204	cathelicidin 1 precursor	502.4
LOC100136187	cathelicidin antimicrobial peptide	234.4
LOC110514021	CD83 antigen-like	15.0
LOC110533206	C-type lectin domain family 4 member M-like	30.2

**Table 4 pathogens-10-01179-t004:** Comparison of the fold change of secreted IgT and IgM transcripts in the intestine of resistant and susceptible steelhead trout at 21 days post exposure to *Ceratonova shasta*.

	Resistant		Susceptible	
	Fold Change	(FDR) *p*-Value	Fold Change	(FDR) *p*-Value
IgT	336.9	1.03 × 10^24^	65.5	2.27 × 10^−8^
IgM	14.5	1.03 × 10^17^	2.3	0.219763

## Data Availability

All sequencing data generated in this study will be made publicly available on the NCBI’s short read archive (SRA) by the time of publication. All other relevant data are contained within the article.

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
