# Peer review of "Intestinal Transcriptomic and Histologic Profiling Reveals Tissue Repair Mechanisms Underlying Resistance to the Parasite Ceratonova shasta"

_pathogens, 2021, doi:10.3390/pathogens10091179_

Round 1

Reviewer 1 Report

The manuscript Pathogens-1331100 “Intestinal Transcriptomic and Histologic Profiling Reveals Tissue Repair Mechanisms Underlying Resistance to the Parasite Ceratonova Shasta” is an elegant work examining the transcriptional changes induced by the myxozoan parasite in rainbow trout with different susceptibility to the infection. The work presented is novel, very difficult to put together, thus I firmly believe it will become a reference for future studies in the field of fish immunology in relation to myxozoan-caused infections. I do believe the manuscript is suitable for publication in Pathogens journal, but after minor revision.

The introduction and methods sessions are well written and provide all the necessary info. The discussion session should be shortened and made more specific by providing more adequate comparisons to other studies available from myxozoan infections in trout.

The work is well carried out, but the presentation of results achieved should be improved. The results session needs some good work to improve the text for the description of retrieved data. In every paragraph it must by specified about the comparisons analyzed, and what was retrieved after data analysis (e.g. paragraph 2.4). Please specify what does it mean "copies" when describing changes in DEGs, using this term it misleading as it looks like they are technical duplicates, so it that case the average should be reported in the tables. Please clarify what kind of entities they are, such as isoforms, analogues. Tables 2 and 3 need a rearrangement and should be supported by an adequate description in each paragraph. The description of results from both DEG and GO analysis is very minimalistic and must be improved.

Suggestions for specific changes, including with some request for specifications, minor text and figure/table changes, are provided in the file attached.

Author Response

We would like to thank the reviewer for the thoughtful and detailed edits they have provided. We have accepted most of the grammatical/minor text changes they have suggested. Additionally, we have adjusted the results section to make it clearer; including specifying which comparisons were analyzed, replaced "copies" with "paralogs", and added more description to the tables.

Reviewer 2 Report

The manuscript presents the transcriptomic and histological profile of immune resistance behind infections caused by the myxozoan parasite Ceratonova shasta. This study adds relevant body of knowledge to the literature related to resistant genes associated with harmful parasites affecting farmed fish such as salmon and trout. The expansion of the aquaculture accompanied by its intensification and global weather changes have predispose farmed fish to more frequent and serious disease outbreaks. This is particularly relevant for parasitic infectious as very few options of treatments are available for their control in fish farms. Therefore, the understanding of resistant genes associated with infectious caused by myxozoans can provide innovative alternatives to manage these parasites within the aquaculture industry.  The manuscript is well written but I suggested some changes and comments (attached document) to improve the quality of the article 

Author Response

We would like the thank the reviewer for their thoughtful edits. We have accepted most of the minor edits they have suggested and made adjustments to figures 2 & 3 to make it clearer which images are being discussed. While we appreciate the comments on the organization of the discussion, we feel the manner it is presented is clear and best presents our ideas so we have retained the original structure.

Reviewer 3 Report

In the current manuscript, the authors analyzed the transcriptomic and histopathologic profile of steelhead fish resistant to the parasite C. shasta at different timepoints following disease challenge. In general, the manuscript is well written, and the results are well presented and discussed. However, I have some comments and suggestions to the authors.

1. Comparison of the transcriptome of both susceptible and resistant fish is interesting. However, these transcriptomes were sequenced and analyzed in two different experiments, maybe different environmental conditions which are expected to affect the gene expression. Although the parasite burden and the sampling timepoints were the same, other environmental factors exist and their effect cannot be excluded without further investigation. A statement regarding this point should be added and clarified so that the reader gets only the conclusions that are supported by the results.

2. The authors did not verify the gene expression results obtained by transcriptomic profiling using qPCR. They should have selected some candidate genes and verified their expression with qPCR.

General comment for the manuscript: Numbers less than 10 should be spelled out. Numbers in the beginning of a sentence should be spelled out.

Methods:

Line 570: Please provide more information on the well water quality.

Line 571: More information about the commercial diet and fish feeding should be provided. For example, the crude protein %, the rate of feeding, … etc.

Lines 589: Please mention the sampling timepoints here.

Line 602: Spell out the number in the beginning of the sentence.

Line 606: What is “Buffer AE”?

Minor comments:

Line 4: Shasta should be lowercase.

Line 121: Ceratonova shasta should be italicized. Also, line 151, 157, 211, 219, … etc. This applies to most figure legends and tables where Ceratonova shasta is mentioned.

Line 215: Define “GO” the first time it is used.

Figure 1: The figure legend should be below the figure.

Lines 580 and 584: Spell out 21 and 5 in the beginning of the sentences.

General comments for the references

The authors should be consistent in citing the references list. Some article names are in “Sentence case” while for others, “Each Word Is Capitalized”.

Also, scientific names should be italicized in the reference list too. Example, Paralichthys olivaceus in number 5, Salmo salar in number 7, … etc.

Reference # 66, 69: citation date 18 Jan 2019??

Author Response

We would like to thank the reviewer for the feedback they have provided us.  We have addressed most of the minor edits in the manuscript. 

"1. Comparison of the transcriptome of both susceptible and resistant fish is interesting...."

We have added a clarifying statement to make it clear that this analysis involves two different studies, and that differences in the experimental setup may lead to changes in gene expression.

"2. The authors did not verify the gene expression results obtained by transcriptomic profiling using qPCR. They should have selected some candidate genes and verified their expression with qPCR."

While we understand this critique, we believe it is unnecessary for several reasons:

  1. qPCR validation is a carry-over from the days of microarrays, when results were less trustworthy. RNA-seq has been in common use for over a decade and it's accuracy and reproducibility is well-established. Because of this, more and more RNA-seq papers are being published without qPCR validation.  https://doi.org/10.1111/mec.15304 https://doi.org/10.1371/journal.pone.0233621 10.1186/s12864-017-3945-6
  2. When RNA-seq and qPCR results are discordant, it is primarily when genes that are lowly expressed (log fold change < 2). Whereas in our manuscript, the genes we specifically talk about are either massively upregulated, have multiple paralogs upregulated, have numerous genes within the pathway upregulated, or some combination of these three, which further supports our finding. 10.1016/j.bioflm.2021.100043
  3. Our samples were deeply sequenced and we have 4 samples per group, rather than the standard 3 samples per group.
  4. Salmonid fish have undergone a relatively recent 4th whole-genome duplication event and because of this they have a very high degree of paralogous genes. While these genes may retain the same function, they may have diverged in their regulation, exhibiting tissue specific expression, or temporal differences in expression. Differentiating paralogs via qPCR is not trivial, and that is part of the reason we chose to use RNA-seq to begin with. https://pubmed.ncbi.nlm.nih.gov/19825389/